# Spatial and radiometric characterization of multi-spectrum satellite images through multifractal analysis

**Carmelo Alonso[1,2], Ana M. Tarquis[2,3], Ignacio Zúñiga[4] and Rosa M. Benito[2]**

[1]Earth Observation Systems, Indra Sistemas S.A., Madrid, Spain.

[2]Grupo de Sistemas Complejos, U.P.M, Madrid, Spain.

[3]CEIGRAM, E.T.S.I.A.A.B., U.P.M, Madrid, Spain.

[4]Dpt. Física Fundamental, Facultad de Ciencias, Universidad Nacional de Educación a Distancia (UNED), Madrid, Spain.

Corresponding author: Ana M. Tarquis (anamaria.tarquis@upm.es)

# Spatial and radiometric characterization of multi-spectrum satellite images through multifractal analysis

**Abstract:** Several studies point out that vegetation indexes can been used to estimate root zone soil moisture and earth surface images, obtained by high resolution satellites, give presently huge information on these indexes based in several wavelengths data. Because of the potential capacity for systematic observations at various scales, remote sensing technology extends possible data archives from present time to over several decades back. For this advantage, enormous efforts have been made by researchers and application specialists to delineate vegetation indexes from local scale to global scale by applying remote sensing imagery.

In this work, four bands images have been considered, involved in these vegetation indexes, taken by satellites Ikonos-2 and Landsat-7 of the same geographic location to study the effect of both spatial (pixel size) and radiometric (number of bits coding the image) resolution on these wavelength bands as well as two vegetation indexes: the Normalized Difference Vegetation Index (NDVI) and the Enhanced Vegetation Index (EVI).

In order to do so, a multifractal analysis of these multi-spectral images was applied in each of these bands and the two indexes derived. The results showed that spatial resolution has a similar scaling effect in the four bands, but radiometric resolution has a larger influence in Blue and Green bands than in Red and Near InfraRed bands. The NDVI showed a higher sensitivity to the radiometric resolution than EVI. Both were equally affected by the spatial resolution.

From both factors, the spatial resolution has a major impact in the multrifractal spectrum for all the bands and the vegetation indexes. This information should be taken in to account when vegetation indexes based on different satellite sensors are obtained.

**Keywords:** vegetation index, wavelengths pattern, multifractal spectrum

## 1 Introduction

Soil moisture is a critical condition affecting interaction of land surface and atmosphere. Remotely sensed data is an important source of information and it can indirectly measure soil moisture in space and time. However, the signal only penetrates the top few centimeters, and soil moisture at deeper layers must be estimated. One method to estimate soil moisture at deeper layers is through vegetation indices. Several authors have investigating the potential of vegetation indices to estimate root zone soil moisture. The Normalized Difference Vegetation Index (NDVI) and Enhanced Vegetation Index (EVI) have been used by several authors (Wang et al., 2007; Ben-Ze'ev et al., 2006; Deng et al., 2007) in different conditions finding significant estimations with root zone soil moisture. For the estimation of these indexes NIR, Red and Blue wavelengths are needed (Huete et al., 2014).

The images provided by the satellites show the land surface in a wide range of wavelengths (from visible to thermal infrared or microwaves) and also with a great variety of spatial resolutions (from a few kilometres to tens of centimetres). The analysis of these varied images and their synergic possibilities are a challenging problem especially with new sensors, which have small spatial resolution and a large range of radiometric quantification. Fractal analysis offers significant potential for improvement in the measurement and analysis of spatially and radiometrically complex remote sensing data. This analysis also provides quantitative insight on the spatial complexity in the information of the landscape contained within these data.

In the general mathematical framework of fractal geometry many analytical methods have been developed; to name a few: textural homogeneity has been characterized using the fractal dimension (Fioravanti, 1994); it has also been used as a spatial measure for describing the complexity of remote sensing imagery (Lam and De Cola, 1993); changes in the image complexity have been detected through the spectral range of hyperspectral images affecting the fractal dimension (Qiu et al. 1999); similarly De Cola (1989) and Lam (1990) have found that fractal dimension also depends on the spectral bands of Landsat-7 TM imagery.

Motivated by the fractal geometry of sets (Mandelbrot, 1983), the development of multifractal theory, introduced in the context of turbulence, has been applied in many areas such as earthquake distribution analysis (Hirata and Imoto, 1991), soil pore characterization (Kravchenko et al. 1999; Tarquis et al. 2003), image analysis (Sánchez et al. 1992) or remote sensing (Tessier et al., 1993; Cheng and Agterberg, 1996; Schmitt et al., 1997; Laferrière and Gaonac'h, 1999; Cheng, 1999; Lovejoy et al., 2001b; Du and Yeo, 2002; Parrinello and Vaughan, 2002; Harvey et al., 2002; Turiel et al. 2005).

The acquisition of remotely sensed multiple spectral images is thus a unique source of data for determining the scale invariant characteristics of the radiance fields related to many factors, such as soil and bedrock chemical composition, humidity content and surface temperature (e.g., Laferrière and Gaonac'h, 1999; Maître and Pinciroli, 1999; Lovejoy et al., 2001a, b; Harvey et al., 2002; Beaulieu and Gaonac'h, 2002; Gaonac'h et al., 2003). In one of the scheme used in the multifractal analysis, the satellite image is considered as a mass distribution of a statistical measure on the space domain studied and it is analyzed through a multifractal (MF) spectrum (Cheng, 2004; Mao-Gui Hu, 2009; Tarquis et al., 2014), which gives either geometrical or probabilistic information about the pixels distribution with the same singularity. Another technique focus its attention in the variations of a measure analyzing the moments of the absolute differences of their values at different scales, the Generalized Structure Function and the Universal Multifractal model (Lovejoy et al., 2001, 2008; Renosh et al., 2015)

The aim of this work is to characterize by MF analysis the image patterns in the wavelength ranges used for NDVI, EVI and Green bands as well as both vegetation indexes. In order to investigate how the image information is affected by the sampling with different spatial and radiometric resolutions, we have also analyzed images of the same site but acquired by two different satellites: Landsat-7 and Ikonos-2.

We present a comparative analysis of multifractal (MF) tools applied to multi-spectral images obtained by Ikonos-2 and LANDSAT-7. Both satellites have several bands in visible and near-infrared spectral regions in common that can be used in vegetation indexes estimation.

However the bands have different spatial resolution, 4 m for Ikonos-2 and 30 m for

LANDSAT-7, and radiometric resolution, 11 and 8 bits respectively. The bands we have

chosen are Red (R), Green (G), Blue (B) and Near InfraRed (NIR). For each of those bands,

the MF spectrum has been calculated directly from the Hölder exponents $\alpha$ and the

singularities spectrum f($\alpha$). The same calculations were applied for NDVI and EVI estimated

on R, B and NIR bands for each image.

## 2    Materials and methods

### 2.1    Images

As already noted, in this work we have analysed two images of the same site acquired from

different satellites, Landsat-7 and Ikonos-2. Both are multi-spectral images with several bands

that cover several regions of the electromagnetic spectrum in the visible and near infrared

wavelength.

Landsat-7 was put in orbits in April 1999. This satellite follows a sun-synchronous orbits at

705 km of altitude, with an equatorial crossing time of 10:00 a.m. in the descending node. It

requires 98.8 min to circle the Earth, tracing a worldwide reference system (WRS) of just

over 230 ground paths. During at least three decades Landsat-7 orbits over each of these paths

once every 16 days in a repetitive cycle (Mika, 1997).

The main Landsat-7 sensor for Earth observation is the Enhanced Thematic Mapper Plus

(ETM+). The ETM+ operates as a whiskbroom scanner and acquires data for seven spectral

bands: visible (ETM+#1, from 0.45 to 0.52 μm; ETM+#2, from 0.53 to 0.61 μm; ETM+#3,

from 0.63 to 0.69 μm), near infrared (ETM+#4, from 0.78 to 0.9 μm), shortwave infrared

(ETM+#5, from 1.55 to 1.75 μm, and ETM+#7, from 2.09 to 2.35 μm) and thermal infrared

(ETM+#6, from 10.4 to 12.5 μm). The ETM+ ground sampling distance (pixel size in the

images) is 30 m for the six reflective bands and 60 m for the thermal band. The ETM+ also

acquires images for a panchromatic band (ETM+#8, from 0.52 to 0.9 μm) with a 15 m ground

sampling distance. The radiometric resolution of the Landsat-7 data is 8 bit/pixel or 256 grey

levels for the pixel digital value.

2   Ikonos-2 was launched in September 1999. Its panchromatic sensor, with a resolution of 0.82

3   m, provided the first very high resolution images of the Earth's surface from EOS. The

Ikonos-2 orbits altitude is approximately 681 km, it is inclined 98.1 degrees to the equator and

it provides sun-synchronous operation. The equatorial crossing time of Ikonos-2 is 10:30 am

in the descending node. The orbits provides daily access to sites within 45 degrees of nadir

(Dial et al. 2003).

The multi-spectral sensor simultaneously collects blue (IK#1, from 0.445 to 0.516 μm), green

(IK#2, from 0.506 to 0.595 μm), red (IK#3, from 0.632 to 0.698 μm) and near infrared (IK#4,

from 0.757 to 0.853 μm) bands with 3.28 meter resolution at nadir. Both images,

panchromatic and multi-spectral, have a radiometric resolution of 11 bits/pixel or 2048 grey

levels for the pixel digital value.

The Landsat-7 multi-spectral image used in this study was acquired on August 6[th] 2000 at

10:46 a.m. and it corresponds to the scene with WRS coordinates, path and row: 201 - 32.

This scene is located in the central region of Spain and it covers a square surface of

approximately 180 km side-length, located around Madrid. Solar azimuth and elevation

angles for this scene are 132.44 and 58.62 degrees respectively.

The Ikonos-2 data used in this study is a multi-spectral image acquired on August 8[th] 2000 at

11:03 a.m. It covers a square area of 11 km side located near Aranjuez, south of Madrid, in

the central region of Spain. Solar azimuth and elevation angles for this scene are 139.5 and

60.79 degrees respectively. Both images were corrected geometrically to the same

cartographic projection: UTM, zone 30 N by a co-registration process.

The analysis has been carried out on a subset that covers (approximately) the same area in

both the Landsat-7 and the Ikonos-2 images, corresponding to a region located north the town

of Aranjuez. The representative elements of the land used in the selected area are: irrigate

crops, pastures, heaths, unirrigated land cultivations and olive groves. The Landsat-7 subset

image is a square of 512 x 512 pixels with a size of 30 m covering a somewhat larger surface than the Ikonos-2 image. The Ikonos-2 image consists of a square subset with 2048 x 2048 pixels and 4 m resolution.

## 2.2 Vegetation indexes

Vegetation is one of the landscape's elements that have received most attention in the field of image analysis. Therefore, there are many parameters that can be used to obtain information on vegetation from remote sensing imagery.

One of the main parameters are the so-called vegetation indexes. These indices allow to detect the presence of vegetation in an area and its activity, since its values are related to this activity. For this, we can use the reflectance values corresponding to the different wavelengths, interpreting these in relation to the photosynthetic activity. Of these indices, the most commonly used is the Normalized Difference Vegetation Index.

The Normalized Difference Vegetation Index is defined by

$$NDVI = \frac{NIR - R}{NIR + R} \tag{1}$$

where NIR is the pixel value in the near infrared band and R the pixel value in the red band. The values of this index are within the range (-1, 1) and their positive values are sensitive to the proportion of soil and vegetation in each pixel (Carlson and Ripley, 1997). Pixels with NDVI < 0.2 are considered without vegetation or bare soil. Pixels with NDVI > 0.5 are considered as fully covered by vegetation.

Other vegetation index is called Enhanced Vegetation Index. It is defined by

$$EVI = 2.5 \frac{(NIR - R)}{(L + NIR + C_1 R - C_2 B)} \tag{2}$$

where NIR is the pixel value in the near infrared band, R the pixel value in the red band and B the pixel value in the blue band. $L$, $C_1$ and $C_2$ are constants with the values 1, 6 and 7.5 respectivelly. The main characteristic of this index is that it corrects some distortions caused by the light dispersion from aerosols, as well as the background soil (Huete et al., 2014).

## 2.3 Multifractal image analysis

A monofractal object can be measured by counting the number N of δ size boxes needed to cover the object. The measure depends on the box size as

$$N(\delta) \propto \delta^{-D_0} \tag{3}$$

where

$$D_0 = \lim_{\delta \to 0} \frac{\log N(\delta)}{\log \frac{1}{\delta}} \tag{4}$$

is the fractal dimension. $D_0$ is calculated from slope of a log-log plot. However, many examples are found where a single scaling law cannot be applied and it is necessary to do a multiscaling analysis.

There are several methods for implementing multifractal analysis. The Universal Multifractal (UM) model assumes that multifractals are generated from a random variable with an exponentiated extreme Levy distribution (Lavallée et al., 1991; Tessier et al., 1993). In UM analysis, the scaling exponent K(q) is highly relevant. This function for the moments q of a cascade conserved process is obtained according to Schertzer and Lovejoy (1987) as follows:

$$K(q) = \begin{cases} \frac{C_1(q^{\alpha_L} - q)}{\alpha_L - 1} & if\ \alpha_L \neq 1 \\ C_1 q \log(q) & if\ \alpha_L = 1 \end{cases} \tag{5}$$

where $C_1$ is the mean intermittency codimension and $\alpha_L$ is the Levy index. These are known as the UM parameters.

Other method is the moment method developed by Halsey et al. (1986) and applied to this case study. This method uses mainly three functions: $\tau(q)$, known as the mass exponent function, $\alpha$, the coarse Hölder exponent, and $f(\alpha)$, multifractal spectrum. A measure (or field), defined in two-dimensional image embedding space ($n \times n$ pixels) and with values based on grey tones (for 8 bits goes from 0 to 255), cannot be consider as a geometrical set and therefore cannot be characterized by a single fractal dimension.

To characterize the scaling property of a variable measured on the spatial domain of the studied, it divides the image into a number of self-similar boxes. Applying disjoint covering by boxes in an "up-scaling" partitioning process we obtain the partition function $\chi(q,\delta)$ (Feder, 1989) defined as:

$$\chi(q,\delta) = \sum_{i=1}^{N(\delta)} \mu_i^q(\delta) = \sum_{i=1}^{N(\delta)} m_i^q \qquad (6)$$

where $m$ is the mass of the measure, $q$ is the mass exponent, $\delta$ is the length size of the box and $N(\delta)$ is the number of boxes in which $m_i > 0$. Based on this, the mass exponent function $\tau(q)$ shows how the moments of the measure scales with the box size:

$$\tau(q) = \lim_{\delta \to 0} \frac{\log < \chi(q,\delta) >}{\log(\delta)} = \lim_{\delta \to 0} \frac{\log < \sum_{i=1}^{N(\delta)} m_i^q >}{\log(\delta)} \qquad (7)$$

where $<>$ represents statistical moment of the measure $\mu_i(\delta)$ defined on a group of non overlapping boxes of the same size partitioning the area studied.

The singularity index, $\alpha$, can be determined by the Legendre transformation of the $\tau(q)$ curve (Halsey, 1986) as:

$$\alpha(q) = \frac{d\tau(q)}{dq} \qquad (8)$$

The number of cells of size $\delta$ with the same $\alpha$, $N_\alpha(\delta)$, is related to the cell size as $N_\alpha(\delta) \propto \delta^{-f(\alpha)}$, where $f(\alpha)$ is a scaling exponent of the cells with common $\alpha$. Parameter $f(\alpha)$ can be calculated as:

$$f(\alpha) = q\alpha(q) - \tau(q) \qquad (9)$$

Multifractal spectrum (MFS) shown as plot of $\alpha$ vs. $f(\alpha)$, quantitatively characterizes variability of the measure studied with asymmetry to the right and left indicating domination of small and large values respectively (Evertsz and Mandelbrot, 1992). There are three characteristic values obtained from MFS, the singularity $\alpha(q)$ values for $q = \{0,1,2\}$. The first value ($\alpha(0)$) corresponds to the maximum of MFS and it is related to the box-counting dimension of the measure support; the second value is related to information or entropy dimension ($\alpha(1)$) and the third with the correlation dimension. The entropy dimension quantifies the degree of disorder present in a distribution. According to Andraud et al. (1994) and Gouyet (1996) a $\alpha(1)$ value close to 2.0 characterizes a system uniformly distributed throughout all scales, whereas a $\alpha(1)$ close to 0 reflects a subset of the scale in which the irregularities are concentrated. These three values will be shown from each calculation of MFS.

The width of the MF spectrum ($\triangle$) indicates overall variability (Tarquis et al., 2001; 2014) and we have split it in two sections. Section I correspond to values $\alpha(q) < \alpha(0)$ or $q > 0$ and

section II to values with $\alpha(q) > \alpha(0)$ or $q < 0$. In section I the amplitude, or semi-width, was calculated with differences $\Delta^+ = \alpha(0) - \alpha(+5)$, and in section II with $\Delta^- = \alpha(-5) - \alpha(0)$.

To study the asymmetry of the multifractal spectrum we have choose the asymmetry index (AI) estimated as (Xie et al., 2010):

$$AI = \frac{\Delta\alpha_L - \Delta\alpha_R}{\Delta\alpha_L + \Delta\alpha_R} \qquad \begin{aligned} \Delta\alpha_L &= \alpha_0 - \alpha_{min} \\ \Delta\alpha_R &= \alpha_{max} - \alpha_0 \end{aligned} \qquad (10)$$

In our case, $\alpha_0$ is the singularity for q=0 or $\alpha(0)$, $\alpha_{min}$ is $\alpha(+5)$ and $\alpha_{max}$ is $\alpha(-5)$.

Therefore, we can rewrite $AI$ as:

$$AI = \frac{\Delta^+ - \Delta^-}{\Delta^+ + \Delta^-} \qquad (11)$$

Expressing $AI$ as equation (11), we can see that it is a normalized index based on the amplitudes $\Delta^+$ and $\Delta^-$.

There are several works relating the UM model and the multifractal formalism based on $\tau(q)$ (Gagnon et al., 2003; Aguado et al., 2014; Morató et al., 2017 among others) through the equations:

$$f(\alpha) = E - c(\gamma); \; \alpha = E - \gamma \qquad (12)$$

$$\tau(q) = E(q - 1) - K(q) \qquad (13)$$

where $E$ is the Euclidean dimension where the measure is embedded, in this case will be $E=2$, and $c(\gamma)$ is the codimension of the singularity of the density of the multifractal measure $\gamma$.

# 3    Results and Discussion

## 3.1    Radiometric influence in the Multifractal spectrum

To study the influence of radiometric resolution on Ikonos-2 image information complexity, the original pixel code (11 bits) has been transform to 8 bits through a rescaling based on minimum and maximum values between 0 and 255, with the aim of preserving the initial histogram shape.

We first discuss the results obtained for the 2048 x 2048 pixels Ikonos-2 image shown in Fig. 1, in bands combination of false colour (IK#4, IK#3, IK#2 bands combination in RGB visualization). In Fig. 2 IK#1, IK#2, IK#3, IK#4 bands histograms are shown. In the right column are histograms with the original radiometric resolution and in the left column the corresponding histograms rescaled to 8 bits. The histograms present a bimodal structure with a narrow peak of low value pixels (dark grey) showing a sharp maximum and a wider peak around a second lower maximum. For bands IK#1, IK#2, IK#3 the narrow peak maximum corresponds to vegetation, mainly irrigate crops, showing strong water absorption. This effect is particularly important in band IK#3. High value pixels (lighter grey) correspond to ground zones with lower vegetation content. However, as vegetation shows high reflectivity in the near infrared, IK#4 band histogram shows a predominance of high values pixels (lighter grey pixels) corresponding to dense vegetation parts. For both radiometric resolutions the shapes of the histograms are very similar as it was our intention (see Fig. 2).

We cover the image with boxes of size $\delta = 2^{-n}$ and we change the box size from 2048 to 2 pixels, that is, $\delta = 2048/2^n$ with $n = 0, 1, 2,..., 10$. For each value of the parameter $q$, from -5 to +5 with increments of 0.5, the partition function (equation 6) is computed and $\log \chi(q,\delta)$ vs $\log\delta$ is plotted in Fig. 3. Each graph contains 11 points and from these a range of scales are selected for the least-square linear fit reaching the maximum possible scales and with a standard error in the slope, the estimated values of $\tau(q)$, less than 0.01. Then, using Eq. [7-8], $\alpha(q)$ and $f(\alpha)$ are obtained. Comparing the range of scales used in both radiometric resolutions, the bands using the original data (11 bits) showed a wider range of scales for the

linear fit, up to 4 pixels, whereas in the 8 bits radiometric resolution were required up to 32 pixels (see arrows in Fig. 3).

The MF spectra $f(\alpha)$ corresponding to the four bands of multispectral Ikonos' images are shown in Fig. 4. These differences found in the multiscaling behaviour of each band are in agreement with previous works (Cheng, 2004; Lovejoy et al., 2008). Just by visual observation, it is remarkable the difference in the bands #3 and #4, R and NIR respectively, between 8 and 11 bits. Higher radiometric resolution gives a higher range of possible grey values per pixel. Note that this radiometric resolution effect is manifested in both sections of the MF spectra (for $q > 0$ and for $q < 0$).

Some characteristic parameters obtained from these MF spectra are shown in Table 1 and Table 2. As expected, in both radiometric resolutions and in each band the $\alpha(0)$ is practically 2, as the measure is defined in the entire plane and it has an Euclidean dimension of 2. Respect to the $\alpha(1)$ value certain differences are found. Comparing the bands in 8bits to the same ones in 11bits always the entropy dimension was higher. However, considering the standard errors only IK#1 (B) and IK#2 (G) bands were significantly different showing the B band the highest difference. Meanwhile, R and NIR bands aren't significantly different. This point out that a more uniformly distribution in space for the bands of Ikonos-2 8bits than in 11bits. The same behaviour is observed in the $\alpha(2)$.

The amplitudes calculated ($\Delta^+$ and $\Delta^-$) in Ikonos-2 11bits bands presents opposite trends (Table 1). Note that amplitude $\Delta^+$ decreases as bands wavelength grows whereas the other amplitude $\Delta^-$ diminishes. Observing these parameters in Ikonos-2 8bits bands (Table 2) a different trend and behaviour are found. In this case both $\Delta^+$ and $\Delta^-$ increase as the

wavelength increases for the three visible bands, but decreasing for the near infrared band (IK#4).

The $AI$ estimated on these MFS amplitudes on each radiometric resolution are showed in the last column of Table 1 and Table 2. Comparing the bands in 8bits to the same ones in 11bits the behaviour is similar; there is a decreasing trend from IK#1 to IK#4, although the range of values is different. At a resolution of 11bits from a positive $AI = 0.240$ at B band goes to a negative $AI = -0.237$ at NIR band. On the other hand, at a resolution of 8bits from an $AI = 0.092$ goes to a negative $AI = -0.347$. The MFS more symmetric are found in G and R bands at a resolution of 11bits and in B and G bands at a resolution of 8bits.

Doing the same study for the vegetation indexes we found the following. The bi-log plot of the partition function ($\chi(q,\delta)$) *vs* $\delta$ is plotted in Fig. 5 for both VI at both radiometric resolutions. Each graph contains 11 points as the bands from where they were estimated. The linear fit was done with the same methodology that for the four bands. In this case only EVI at 8bits show a better linear trend in a wider range of scales. However, to better compare both VIs, from both radiometric resolutions, a range achieving 32 pixels (128 m.) was selected as showed by arrows in Fig. 5.

The MF spectra $f(\alpha)$ of EVI and NDVI estimated for both radiometric resolutions of Ikonos' images are shown in Fig. 4. Both vegetation indexes show differences due to the transformation from 11bits to 8bits. However, NDVI shows higher differences in the MFS, mainly in the part corresponding to q negative values (right side). Even EVI presents changes; its MFS is closer at both radiometric resolutions. Comparing the range of $f(\alpha)$ values in the VIs to the range obtained in the four bands (left column in Fig. 4) there is a remarkable contrast. Meanwhile the NIR band of 8bits achieves $f(\alpha)$ value close to 0.5, EVI and NDVI achieve values closer to 0.2. These differences are higher in 11bits image; R band achieves a $f(\alpha)$ value close to 0.9 and VIs achieve again values around to 0.2. The same characteristic

parameters obtained from the bands MF spectra were calculated for the vegetation indexes and are shown in Table 1 and Table 2.

In respect to the $\alpha(1)$ values certain differences are found between the vegetation indexes. Comparing the NDVI in 8bits to the same ones in 11bits always the entropy dimension was higher as it was found in the bands. However, EVI shows the contrary, entropy values of 11bits image are higher than 8bits image, although the differences aren't significant. Therefore, the radiometric resolution is affecting more to NDVI than to EVI. The former presents a more uniformly space distribution in 8bits than in 11bits image. The same behaviour is observed in $\alpha(2)$.

The amplitudes calculated ($\Delta^+$ and $\Delta^-$) in Ikonos-2 11bits VIs presents a similar situation (Table 1). The amplitude $\Delta^+$ is lower than amplitude $\Delta^-$ and therefore the $AI$ estimated is negative. This is visually perceived in Fig. 4, right column. Observing these parameters in Ikonos-2 8bits vegetation indexes (Table 2) similar situation are found but the values are lower. In both images (11bits and 8bits) NDVI shower higher values for both amplitudes, $\Delta^+$ and $\Delta^-$.

All the $AI$ estimated for both vegetation indexes on each radiometric resolution are negative (Table 1 and Table 2) indicating a high asymmetry on the right part of the MFS as showed in Fig. 4. Comparing the $AI$ values in 8bits to the same ones in 11bits they are similar pointing out that the shape of the MFS is similar as this index is a normalized index. However the values of the amplitudes mark a higher change in NDVI than in EVI.

## 3.2 Spatial resolution influence in the Multifractal spectrum

A comparison between Landsat, with an original pixel code of 8 bits, and the rescaled histograms from Ikonos, with an original pixel code of 11 bits, is made. In this section, we discuss the results obtained in the MF analysis on 512 x 512 pixels Landsat-7 image shown in Fig. 6, in bands combination of false colour (ETM+#3, ETM+#2 and ETM+#1 bands combination in RGB visualization).

In the right column of Fig. 6, the histograms of the Landsat-7 image for the first four bands are showed. The histograms present a bimodal structure except for ETM+#4 (NIR) which there is only one peak. Comparing these histograms with the ones obtained for Ikonos-2 8bits (Fig. 2) the peaks aren't so abrupt and narrow. At the same time, ETM+#1, ETM+#2 and ETM+#3 bands show the absolute maximum peak at high value pixels (light grey) and a second one at lower value pixels (dark grey). These bands are more centred and don't show a shift to the left as the Ikonos-2 8bits bands (IK#1, IK#2 and IK#3. In the case of NIR band, Landsat-7 and Ikonos-2 8bits are quite similar except for the absence of a second peak.

In the calculations, box sizes range from 512 to 2 pixel, that is, $\delta = 512 / 2^n$ with $n = 0, 1,...8$. For each value of the parameters $q$, from -5 to +5 with increments of 0.5, we compute the partition function, and the bi-log $\chi(q,\delta)$ vs $\delta$ are plotted in Fig. 7. In this case each linear fits contains only 9 points as the size of the image is 512x512 pixels. The same method was applied to select the range of scales used in the linear fit, achieving a scale of 4 pixels. Changing from pixels to meters, the scale achieved used in Landsat-7, in the MF analysis, was around 120 m. In the case of Ikonos-2 8bits the scale was 32 pixels or 128 m, very close to Landsat-7.

The MF spectra, $f(\alpha)$, corresponding to the first four bands of multispectral Landsat-7 images are shown in Fig. 8. From a comparison of figures 4 and 8 we see that Landsat-7 images MF spectra are always located inside the corresponding Ikonos-2 MF spectra. For a given value of Hölder exponent $\alpha$ the relation $f_{Landsat}(\alpha) \le f_{Ikonos}(\alpha)$ is always satisfied. This result means that Landsat-7 images show lower complexity than Ikonos-2 8bits images. As

stated in section 2.1 Ikonos-2 satellite data are coded in 11 bits in contrast with Landsat-7 8

bits coded data. To compare both sensors, with different spatial resolution, we pass Ikonos-2

from 11bits to 8bits observing that the later shows more complexity than Landsat.

The MF spectra parameters from Lansat-7 are shown in Table 3. We will compare in this

section with Ikonos-2 8bits (Table 2). The $\alpha(1)$ values from the four bands of Landsat-7 are

higher than the ones presented by Ikonos-2 8bits indicating a higher uniformly space

distribution. Comparing between the bands, there are not significant differences contrary with

the trend we observed among them in Ikonos-2 8bits. The $\alpha(2)$ shows the same behaviour.

The amplitudes calculated ($\Delta^+$ and $\Delta^-$) in Landsat-7 bands present few variations (Table 3).

The amplitude $\Delta^+$ decreases from ETM+#1 to ETM+#3 and then presents an increase in

ETM+#4 (NIR) whereas the other amplitude $\Delta^-$ remain practically constant. Observing these

parameters in Ikonos-2 8bits bands (Table 2) there are variations in value and behaviour for

the four bands. In this case, both $\Delta^+$ and $\Delta^-$ increase as the wavelength increases for the three

visible bands, but decreasing for the near infrared band (IK#4).

The $AI$ estimated on these MFS amplitudes on each Landsat-7 bands are positive except for

ETM+#3 (R band). For the G band (ETM+#3) the symmetry of the MFS is complete. The

band that shows certain asymmetry is the B band (ETM+#1).

Regarding the vegetation indexes, estimated on Landsat-7 bands, we found the following. The

bi-log plot of the partition function ($\chi(q,\delta)$) *vs* $\delta$ is plotted in Fig. 9 for both VIs. Each graph

contains 9 points as the bands from where they were estimated. The linear fit was done with

the same methodology that for the four bands. EVI and NDVI show the same behaviour and

the same range of scale was selected achieving 8 pixels as showed by arrows in Fig. 9.

The MF spectra $f(\alpha)$ of EVI and NDVI estimated based on Landsat-7 image are shown in Fig. 8. Both vegetation indexes show differences mainly in the right side of the MFS (for q negative values). Comparing the range of $f(\alpha)$ values in the VIs to the range obtained in the four bands (left column in Fig. 8) there is a remarkable contrast. Meanwhile the NIR band of 8bits achieves $f(\alpha)$ value close to 1.6, EVI and NDVI achieve values closer to 1. A similar situation was found with both images of Ikonos-2.

We are going to study the parameters obtained from the MF spectra for the vegetation indexes (Table 3). The results are quite similar to those commented about the Landsat-7 bands showing even higher values, 1.996 NDVI and 1.997 EVI.

The amplitude $\Delta^+$ is quite low compared with the bands and to the VIs of Ikonos-2 8bits. On the other hand, the amplitude $\Delta^-$ is higher than Landsat-7 bands but only a third of the values showed by Ikonos-2 8bits VIs (Table 2). The $AI$ estimated for both vegetation indexes are negative indicating a high asymmetry on the right part of the MFS as showed in Fig. 8. Comparing the $AI$ values of Landsat-7 VIs with the ones of both Ikonos-2 images, these are the highest indicating that show the most unbalance MFS shifted totally on the right side of the spectrum.

**4    Conclusions**

In this work, we have used MF spectra as a successful technique for analyzing common information contained in multi-spectral images of the site of the Earth surface acquired by two satellites, Landsat-7 and Ikonos, in four common bands in the visible (Blue, Green and Red) and near-infrared (NIR) wavelength regions used in several vegetation indexes.

The radiometric resolution has been studied comparing MF spectra of the images acquired by Ikonos-2 coded in 11 bits and transformed in 8 bits code. The results obtained after the histogram transformation in the blue and green bands were the ones you would expected after the simplification applied from 11 to 8 bits, i.e. higher frequency in all the histogram bin values (see Fig. 2). In contrast, red and infrared bands showed no sensitivity at all to this transformation keeping similar MF spectra. To our knowledge, this is the first time these differences among bands are reported.

In order to analyse the effect of spatial resolution in each band at 4 m (Ikonos-2 with 8 bits) pixel size and 30 m (Landsat-7 with 8 bits) pixel size are compared. Obviously, the higher the spatial resolution, the higher the Hölder spectrum amplitude in the green and blue bands are. In fact, observing the graphics of the three cases studied (Ikonos-2 11 bits, Ikonos-2 8 bits and Landsat-7 8 bits) both bands gradually reduce their $\alpha(q)$ amplitude in the negative as well as in the positive q values. However, this is not the case for R and NIR bands that present a much higher difference between Ikonos-2 8bits and Landsat-7 curves of the MF spectra than between Ikonos-2 11bits and 8bits.

In the $q > 0$ MFS region for B and G bands the sensitivity to both factors are very similar, being the B band ratio slightly higher. In the other two bands, R and NIR, for the same region mainly present sensitivity to spatial resolution, showing a similar rate than blue and green bands. Observing the $q < 0$ region for blue and green the behaviour is similar to the positive one but with a lower ratio (between 1 and 2) and once more, the red and infrared bands show slightly sensitivity to radiometric resolution. Nevertheless in the spatial resolution the R band has a ratio similar to B and G, and NIR shows the highest ratio (around 8) pointing the extreme influence of the lowest values contained, see histograms in Fig. 2 (Ikonos-2 8 and 11 bits) and Fig. 6 (Landsat-7).

The implications of these variations in the B, R and NIR in the multi-scaling behaviour of two vegetation indexes, NDVI and EVI, have been also studied. The radiometric resolution showed a higher influence in the MFS of the NDVI than in EVI. This implies that the use of

the B band in the later has a steady effect in the scaling behaviour. As commented for the bands, the spatial resolution had a major impact in both vegetation indexes.

Further research will be conducted to establish a qualitative and quantitative comparison of these conclusions among several multifractal methodologies applied on these images.

## Acknowledgements

Thanks are due to the anonymous referees and the editor for their interest and patient in this work. Discussion and comments suggested by Prof. Jose Manuel Redondo are highly appreciated. This work has been supported by the Ministerio de Economía y Competitividad (MINECO) under Contract Nos. MTM2012-39101 and MTM2015-63914-P.

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

Table 1. Parameters obtained from the multifractal spectrum from each band of Ikonos-2
image, and the vegetation indexes (VI) estimated, with a pixel size of 4 m and a radiometric
resolution of 11 bits. . The amplitudes of $\alpha$ values are presented as $\Delta^+$ and $\Delta^-$ corresponding to
$\alpha(0)$- $\alpha(5)$ and $\alpha(-5)$- $\alpha(0)$ respectively. And the asymmetry index (AI) corresponding to
$\frac{\Delta^+ - \Delta^-}{\Delta^+ + \Delta^-}$.

| | Band | q | $\alpha(q)$ | $\Delta^+$ | $\Delta^-$ | AI |
|---|---|---|---|---|---|---|
| Ikonos-2 (11bits) | IK#1 | 0 | 2.001±0.001 | | | |
| | | 1 | 1.938±0.005 | 0.418 | 0.256 | 0.240 |
| | | 2 | 1.865±0.009 | | | |
| | IK#2 | 0 | 2.001±0.001 | | | |
| | | 1 | 1.936±0.005 | 0.377 | 0.313 | 0.093 |
| | | 2 | 1.871±0.007 | | | |
| | IK#3 | 0 | 2.001±0.001 | | | |
| | | 1 | 1.937±0.005 | 0.348 | 0.382 | -0.047 |
| | | 2 | 1.878±0.006 | | | |
| | IK#4 | 0 | 2.001±0.001 | | | |
| | | 1 | 1.959±0.005 | 0.290 | 0.470 | -0.237 |
| | | 2 | 1.908±0.009 | | | |

| | VI | q | $\alpha(q)$ | $\Delta^+$ | $\Delta^-$ | AI |
|---|---|---|---|---|---|---|
| Ikonos-2 (11bits) | NDVI | 0 | 2.000±0.001 | | | |
| | | 1 | 1.886±0.008 | 0.516 | 1.166 | -0.386 |
| | | 2 | 1.779±0.010 | | | |
| | EVI | 0 | 2.000±0.001 | | | |
| | | 1 | 1.948±0.002 | 0.270 | 0.877 | -0.533 |
| | | 2 | 1.897±0.004 | | | |

Table 2. Parameters obtained from the multifractal spectrum from each band of Ikonos-2

image, and the vegetation indexes (VI) estimated, with a pixel size of 4 m and a radiometric

resolution of 8 bits. The amplitudes of $\alpha$ values are presented as $\Delta^+$ and $\Delta^-$ corresponding to

$\alpha(0)$- $\alpha(5)$ and $\alpha(-5)$- $\alpha(0)$ respectively. And the asymmetry index (AI) corresponding to

$\frac{\Delta^+ - \Delta^-}{\Delta^+ + \Delta^-}$.

| | Band | q | $\alpha(q)$ | $\Delta^+$ | $\Delta^-$ | AI |
|---|---|---|---|---|---|---|
| Ikonos-2 (8bits) | IK#1 | 0 | 2.000±0.001 | | | |
| | | 1 | 1.971±0.003 | 0.231 | 0.192 | 0.092 |
| | | 2 | 1.930±0.006 | | | |
| | IK#2 | 0 | 2.000±0.001 | | | |
| | | 1 | 1.963±0.004 | 0.270 | 0.287 | -0.031 |
| | | 2 | 1.914±0.006 | | | |
| | IK#3 | 0 | 2.000±0.001 | | | |
| | | 1 | 1.945±0.005 | 0.323 | 0.614 | -0.311 |
| | | 2 | 1.887±0.006 | | | |
| | IK#4 | 0 | 2.000±0.001 | | | |
| | | 1 | 1.966±0.004 | 0.248 | 0.512 | -0.347 |
| | | 2 | 1.923±0.008 | | | |

| | VI | q | $\alpha(q)$ | $\Delta^+$ | $\Delta^-$ | AI |
|---|---|---|---|---|---|---|
| Ikonos-2 (8bits) | NDVI | 0 | 2.000±0.002 | | | |
| | | 1 | 1.932±0.005 | 0.337 | 0.984 | -0.490 |
| | | 2 | 1.855±0.008 | | | |
| | EVI | 0 | 2.000±0.002 | | | |
| | | 1 | 1.940±0.004 | 0.300 | 0.874 | -0.488 |
| | | 2 | 1.873±0.006 | | | |

Table 3. Parameters obtained from the multifractal spectrum from each band of Landsat-7

image, ==and the vegetation indexes (VI) estimated,== with a pixel size of 4 m and a radiometric

resolution of 8 bits. The amplitudes of $\alpha$ values are presented as $\Delta^+$ and $\Delta^-$ corresponding to

$\alpha(0)$- $\alpha(5)$ and $\alpha(-5)$- $\alpha(0)$ respectively. ==And the asymmetry index (AI) corresponding to==

==$\frac{\Delta^+ - \Delta^-}{\Delta^+ + \Delta^-}$.==

| | Band | q | $\alpha(q)$ | $\Delta^+$ | $\Delta^-$ | AI |
|---|---|---|---|---|---|---|
| Landsat-7 | ETM+#1 | 0 | 2.001±0.001 | | | |
| | | 1 | 1.985±0.005 | 0.160 | 0.119 | 0.147 |
| | | 2 | 1.960±0.010 | | | |
| | ETM+#2 | 0 | 2.003±0.001 | | | |
| | | 1 | 1.988±0.004 | 0.119 | 0.119 | 0.000 |
| | | 2 | 1.970±0.008 | | | |
| | ETM+#3 | 0 | 2.001±0.001 | | | |
| | | 1 | 1.989±0.004 | 0.095 | 0.110 | -0.073 |
| | | 2 | 1.974±0.007 | | | |
| | ETM+#4 | 0 | 2.017±0.001 | | | |
| | | 1 | 1.989±0.004 | 0.106 | 0.104 | 0.010 |
| | | 2 | 1.973±0.008 | | | |

| | VI | q | $\alpha(q)$ | $\Delta^+$ | $\Delta^-$ | AI |
|---|---|---|---|---|---|---|
| Landsat-7 | NDVI | 0 | 2.001±0.001 | | | |
| | | 1 | 1.996±0.001 | 0.028 | 0.353 | -0.852 |
| | | 2 | 1.992±0.001 | | | |
| | EVI | 0 | 2.001±0.001 | | | |
| | | 1 | 1.997±0.001 | 0.022 | 0.288 | -0.859 |
| | | 2 | 1.994±0.001 | | | |

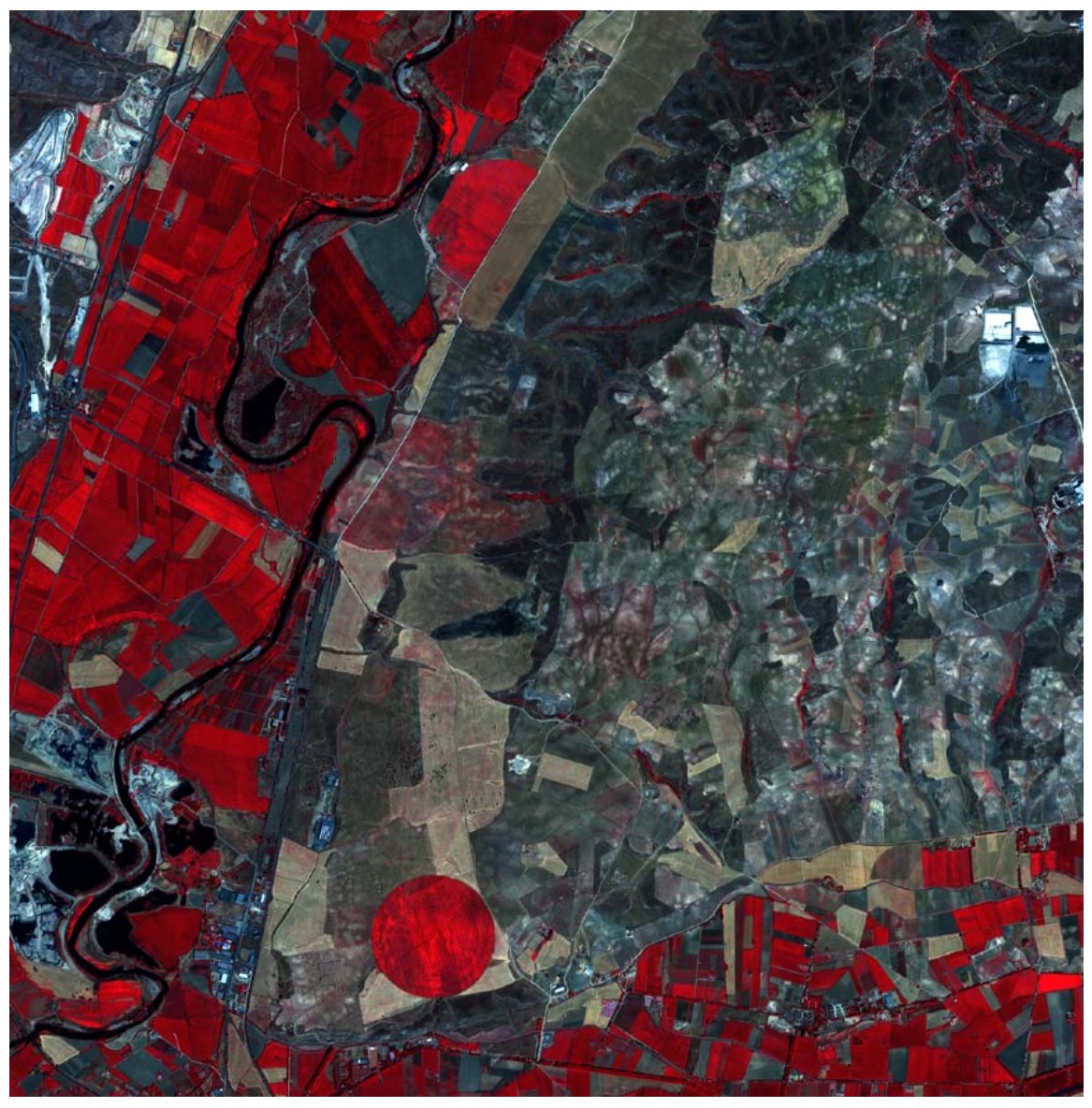

3  Figure 1. The Ikonos-2 image in band combinations of false colour (IK#4, IK#3 and IK#2 in

4  RGB). The image has a size of 2048 x 2048 pixels, each area unit correspond to 4x4 m. The

5  coordinates UTM (zone 30) of the upper left and low right pixel in the image are: ULX =

6  446037 m, ULY = 4441684 m, LRX = 454229 m and LRY = 4433492 m.

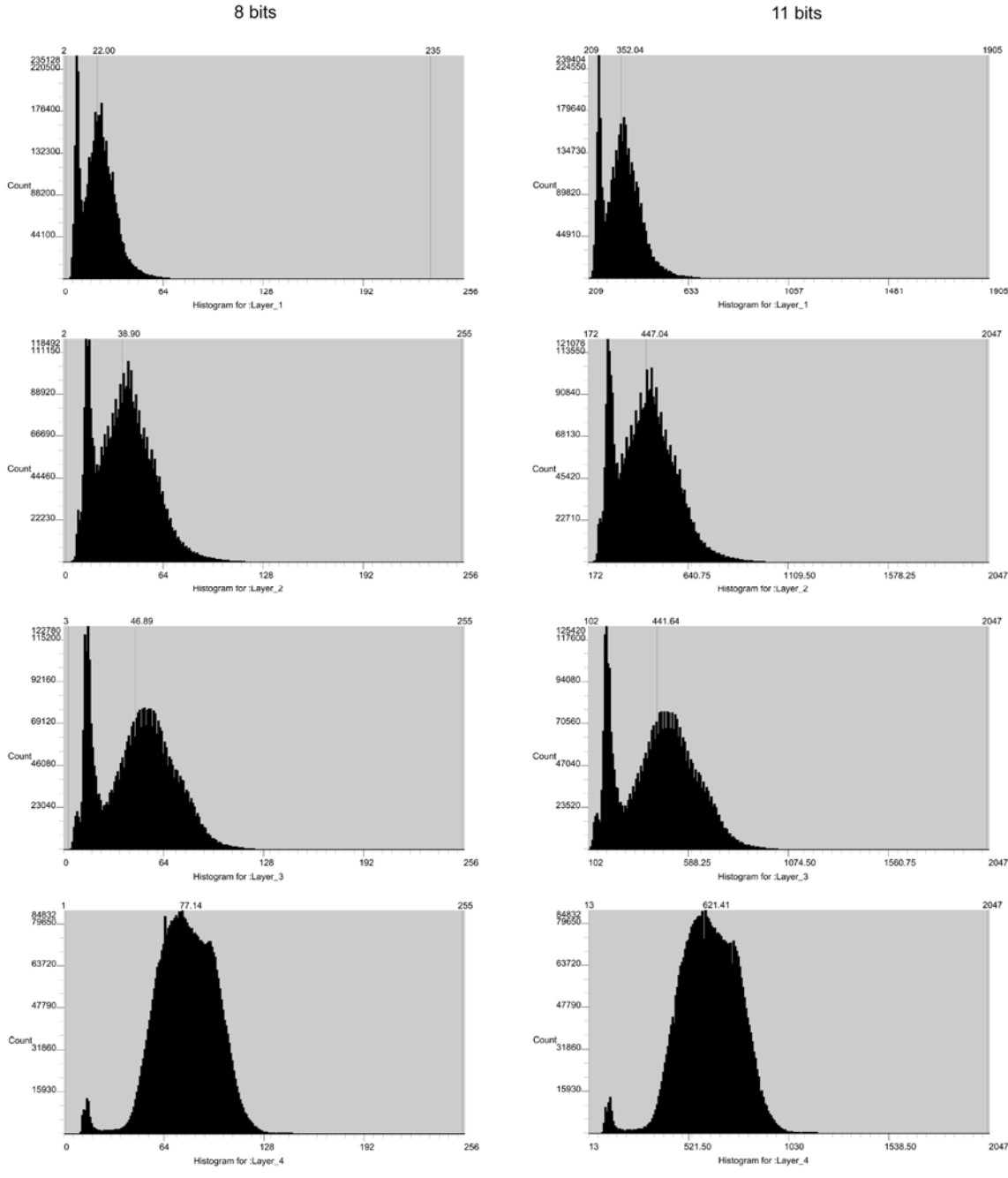

Figure 2. Histograms of the four bands of Ikonos-2 image for the original radiometric resolution, 11 bits (right), and the minimum-maximum rescale 8 bits radiometric resolution (left).

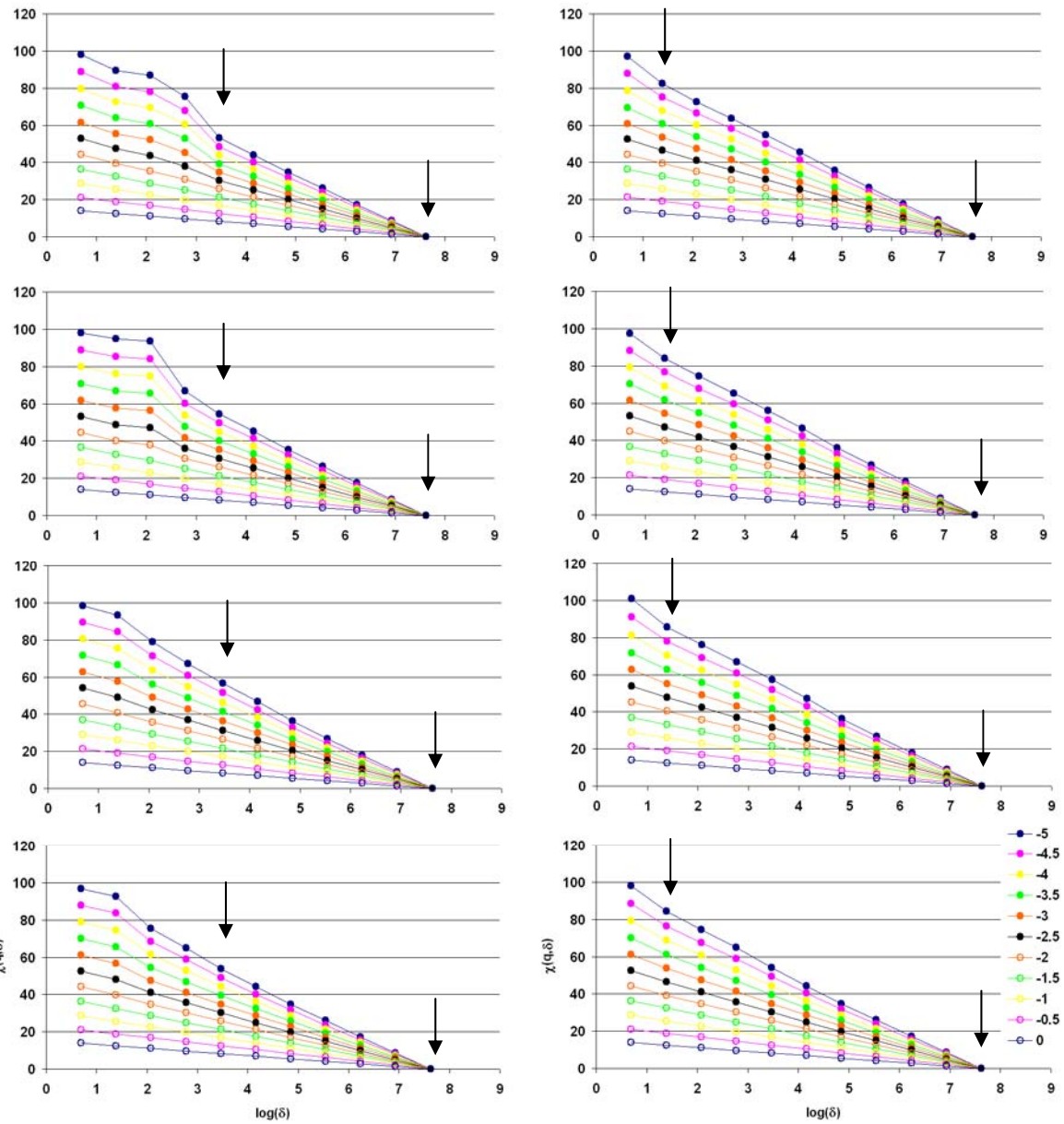

Figure 3. Bi-log plots of the partition function $\chi(q,\delta)$ versus $\delta$ for the first four bands of

Ikonos-2 satellite and for $q < 0$ values. From top to bottom we show the results for IK#1,

IK#2, IK#3 and IK#4. The left column correspond to 8-bits image and the right column to 11

bits image. The arrows marked the range of scales used for the fit and to calculate the slope

for different values of $q$ (7 points in the left column and 10 points in the right column).

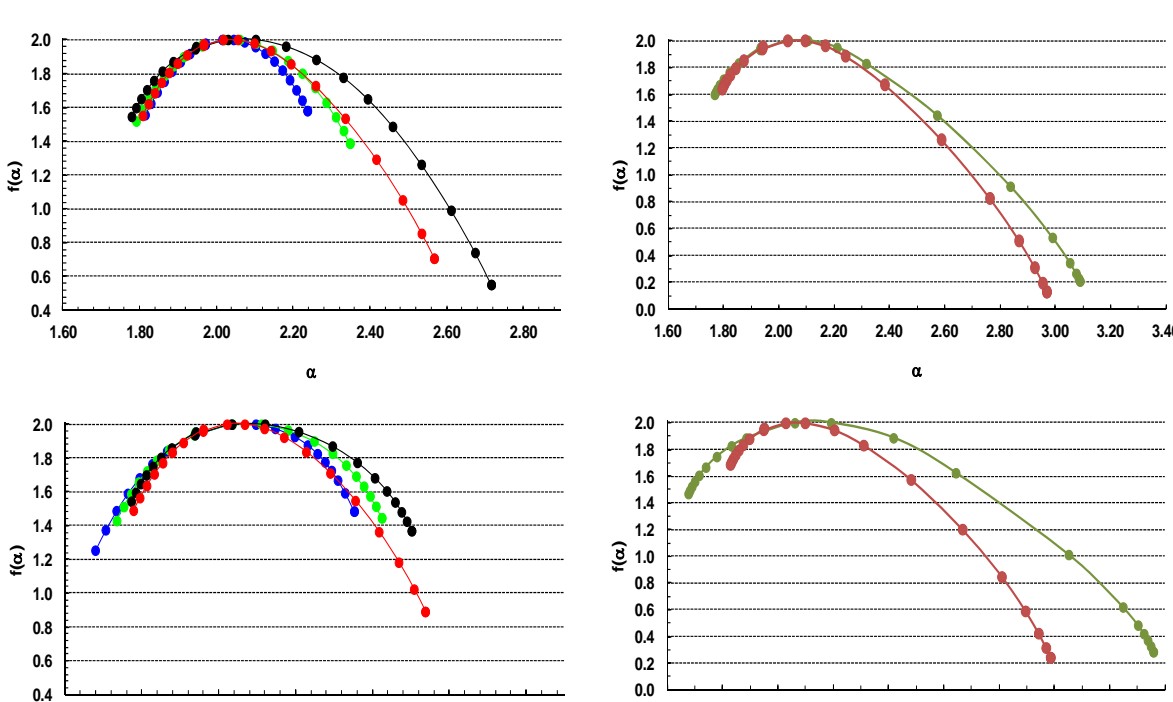

Figure 4. Multifractal spectrum of Ikonos-2 images for the original pixel values coded in 11
bits (lower) and the min-max rescale to 8 bits (upper). Left column correspond to each band
analyzed: IK#1 in blue colour, IK#2 in green colour, IK#3 in red colour and IK#4 in black.
Right column correspond to vegetation indexes: NDVI in green colour and EVI in brown.

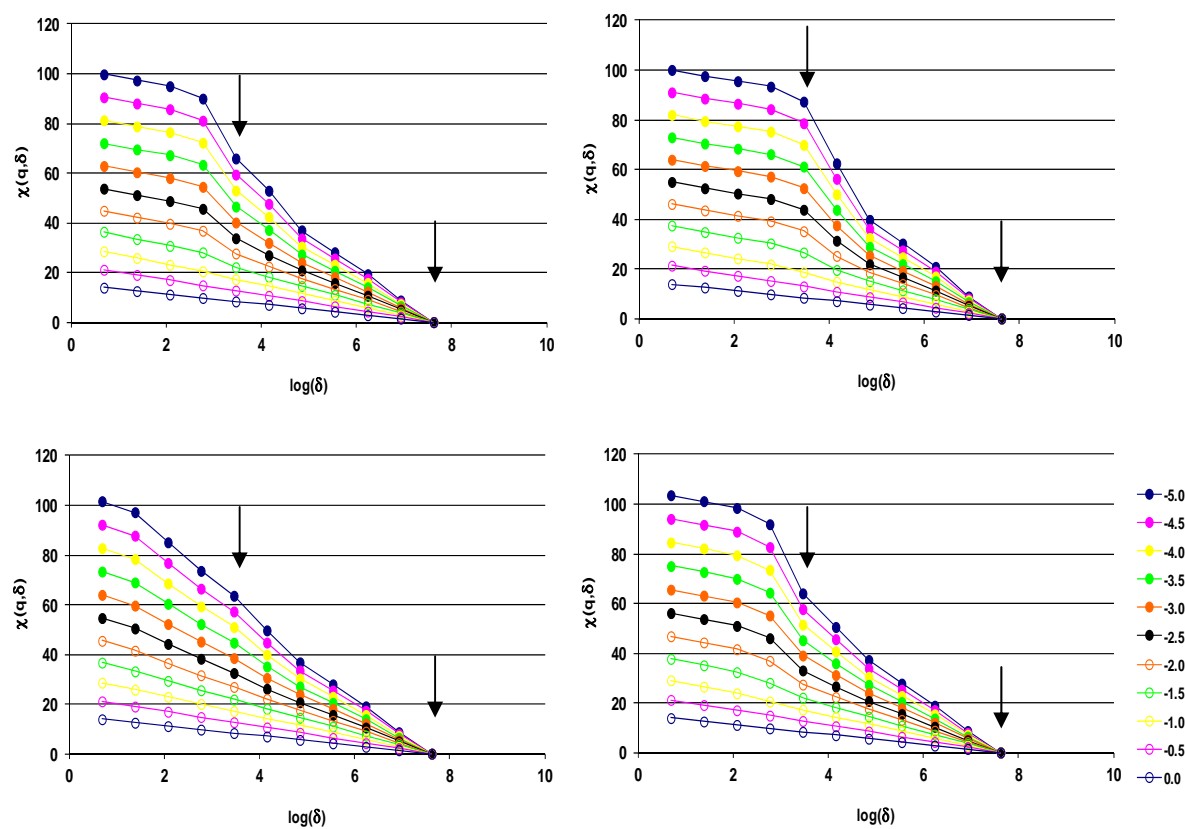

Figure 5. Bi-log plots of the partition function $\chi(q,\delta)$ versus δ for the vegetation indexes estimated from B, R and NIR bands of Ikonos-2 satellite and for $q<0$ values. From top to bottom we show the results for NDVI and EVI respectively. The left column correspond to 8-bits image and the right column to 11 bits image. The arrows marked the range of scales used for the fit and to calculate the slope for different values of $q$ (7 points).

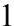

Figure 6. The Landsat-7 image and the histograms for the first four bands: blue (ETM+ #1), green (ETM+ #2), red (ETM+ #3) and near infrared (ETM+ #4). The image has a size of 512x512 pixels, each area unit correspond to 30x30 m. The coordinates UTM (zone 30) of the upper left and low right pixel in the image are: ULX = 442185 m, ULY = 4445568 m, LRX = 457545 m and LRY = 4430208 m.

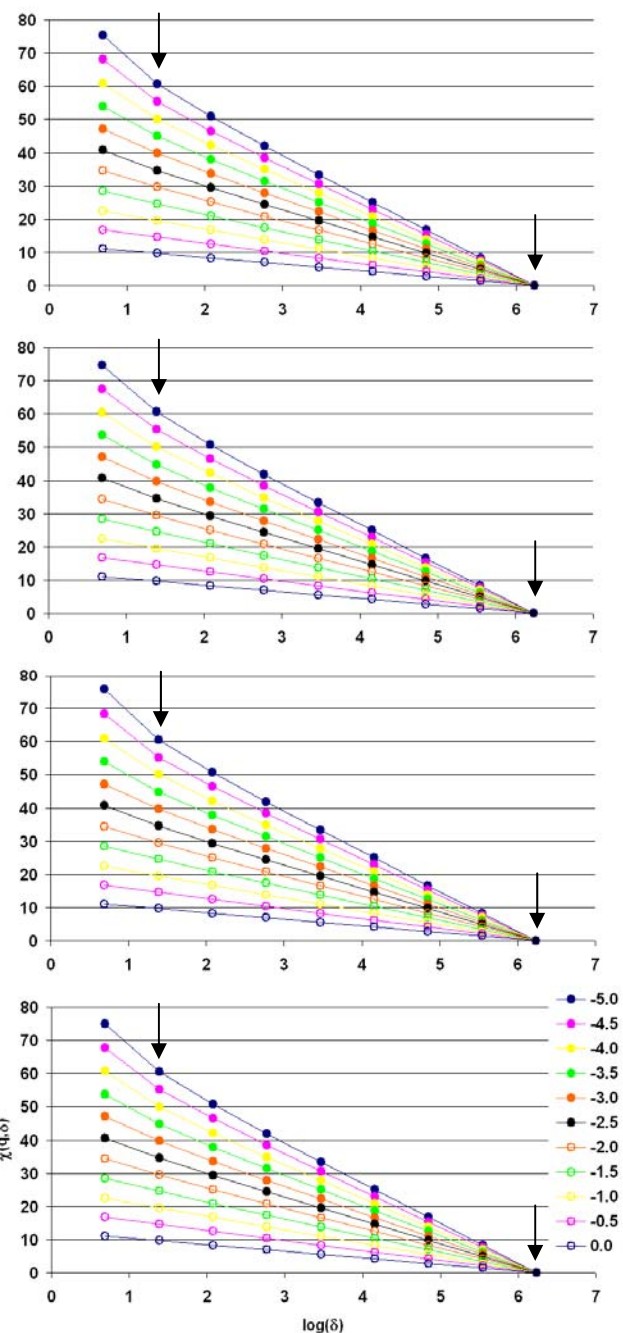

Figure 7. Bi-log plots of the partition function $\chi(q,\delta)$ versus $\delta$ for the first four bands of
Landsat-7 satellite and for $q < 0$ values. From top to bottom we show the results for
ETM+#1, ETM+#2, ETM+#3 and ETM+#4. The arrows marked the range of scales used for
the fit and to calculate the slope for different values of $q$ (8 points).

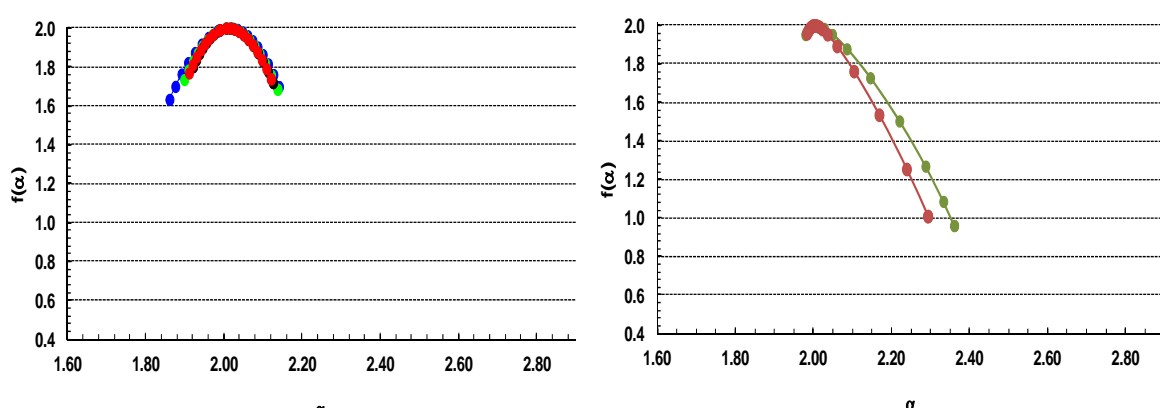

4 Figure 8.  Multifractal spectrum of Landsat-7 image for the original pixel values coded in 8

5 bits. Left plot correspond to each band analyzed: ETM+#1 in blue colour, ETM+#2 in green

6 colour, ETM+#3 in red colour and ETM+#4 in black. Right plot correspond to vegetation

7 indexes: NDVI in green colour and EVI in brown.

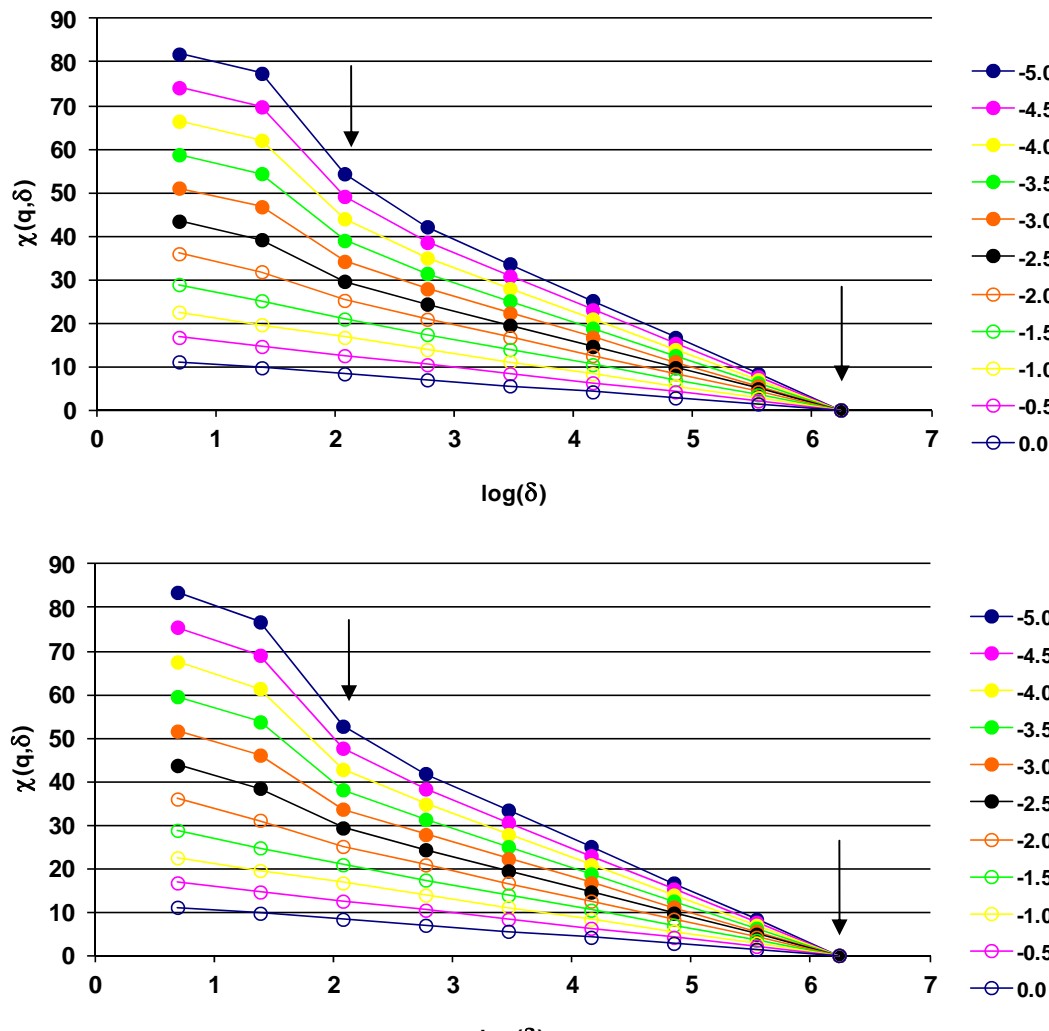

4  Figure 9. Bi-log plots of the partition function $\chi(q,\delta)$ versus $\delta$ for the vegetation indexes

5  estimated from B, R and NIR bands of Landsat-7 satellite and for $q<0$ values. From top to

6  bottom we show the results for NDVI and EVI. The arrows marked the range of scales used

7  for the fit and to calculate the slope for different values of $q$ (7 points).