# Peer review of "Spatial and radiometric characterization of multi-spectrum"

_Nonlinear Processes in Geophysics, 2016_

## Short Comment (SC1) · 5 Oct 2016

In my opinion this is a novel work related to multiscaling analysis of data cropped from satellite images. I would like to pay attention to the following: a)In order to better illustrate similitude or differences in the visible (blue, green and red) and near-infrared wavelength I suggest to characterize assymmetry of the singularity spectrum by AI index (Xie et al., 2010), defined as: AI = (ïĄĎïĄąL-ïĂăïĄĎïĄąR)/(ïĄĎïĄąL+ïĂăïĄĎïĄąR), where ïĄĎïĄąL= (ïĄąïĂř - ïĄąmin) and ïĄĎïĄąR= (ïĄąmaxïĂ∎ïĄąïĂř) are the widths of the left and right branches of the f(ïĄą)-ïĄąïĂăplots, respectively. Reference: Xie, S., Q. Cheng, X. Xing, Z. Bao, and Z. Chen. 2010. Geochemical multifractal distribution patterns in sediments from ordered streams. Geoderma 160:36-46. b) I wonder if it

would be worth checking multifractality of the Normalized Difference Vegetation Index (NDVI)

PLEASE, SEE ALSO ATTACHED TEXT

[Figure]

In my opinion this is a novel work related to multiscaling analysis of data cropped from satellite images. I would like to pay attention to the following:

a) In order to better illustrate similitude or differences in the visible (blue, green and red) and near-infrared wavelength I suggest to characterize assymmetry of the singularity spectrum by AI index (Xie et al., 2010), defined as: $AI = (\Delta\alpha_L - \Delta\alpha_R)/(\Delta\alpha_L + \Delta\alpha_R)$, where $\Delta\alpha_L = (\alpha_0 - \alpha_{min})$ and $\Delta\alpha_R = (\alpha_{max} - \alpha_0)$ are the widths of the left and right branches of the $f(\alpha)-\alpha$ **plots, respectively.**
Reference: Xie, S., Q. Cheng, X. Xing, Z. Bao, and Z. Chen. 2010. Geochemical multifractal distribution patterns in sediments from ordered streams. Geoderma 160:36-46.

b) I wonder if it would be worth checking multifractality of the Normalized Difference Vegetation Index (NDVI)

**Fig. 1.**

---

## Referee Comment (RC1) · Anonymous Referee #1 · 13 Oct 2016

This manuscript characterizes several bands of multi-spectral images obtained with different radiometric and spatial resolution by IKONOS-2 and LANDSAT-7 satellites by multifractal analysis. Three of the studied bands were in the visible wavelength spectrum (red, green and blue), while the fourth one was in the near-infrared region. IKONOS-2 images were taken with a radiometric resolution of 11 bits/pixel and the pixel size was 4m x 4m pixel size. LANDSAT-7 images were taken with a radiometric resolution of 8 bits/pixel and the pixel size was 30m x 30m. For IKONOS-2 images both, the original pixel code (11 bits) and a transformed pixel code (8 bits) have been considered. Then, the effects of spatial and radiometric resolution on several multifractal parameters were investigated. The rationale and the objectives exposed in the

Introduction section are worthwhile and in general the work appears well justified. The main findings are the usefulness of multifractal parameters to: 1) assess different patterns of scaling heterogeneity and evenness of the studied bands, and 2) discriminate between bands with different spatial and radiometric resolution. In general, the paper is well written and organized, and represents an original contribution. The results are based in robust data analysis. This study also is compatible with the aims of Non-linear Processes in Geophysics (NPG) and may fit well into the scope of the current special issue titled "Multifractal Analysis in Soil Systems". In my opinion, it should be acceptable for publication following minor revisions. Some specific comments are next provided. - The IKONOS-2 image presented and analyzed for multifractality is square and consists of 2048 x 2048 pixels of 4 x 4 m. The original LANDSAT-7 image is rectangular and consists of 772 x 828 pixels of 30 x 30 m; however multifractal analysis was not performed on the rectangular image, but on a square of 512 x 512 pixels. In my opinion this should be considered. To be consistent, I suggest modifying Figure 5 for including only the portion that has been used for multifractal analysis. - In general the discussion section should be tightened up. The authors should put an emphasis on what are the novel results and the novel things they have learned using multifractal analysis. - Specifically, I wonder if the authors could include data about the effect of spatial and radiometric resolution on the vegetation indices mentioned in the Introduction, i.e. NDVI (normalized difference vegetative index) and EVI (enhanced vegetation index). Are changes in multifractal parameters related changes in NDVI, EVI or to other physical properties measured or derived from multispectral images? - Tables 1, 2 and 3 list Hölder exponent parameters obtained from singularity spectra for q=1, q=2 and q=3. Please provide also the respective errors of these values. - I disagree with the statement in Page 8, Lines 13 and 14: "In order to avoid any other effect beside the spatial resolution a comparison between Landsat (with an original pixel code of 8 bits) and the rescaled histograms from Ikonos is made". This is because there are other factors of variation including 1) the total size of the image analyzed and 2) difference in the wavelength of a given spectral band between satellite images. Regarding my first

remark, please note that if you change the size of the image, proportions of different vegetation and soil types are also changing and this has effects in the multispectral results. Regarding to the second remark, please note for example that wavelength for ETM#3 of Landsat was from 0.53 to 0.61 micrometers, while those for the same band of Ikonos was from 0.506 to 0.595. - Even if the paper is reasonably well written, as before stated, the English language should be improved before acceptation.
* * *

---

## Referee Comment (RC2) · Anonymous Referee #2 · 8 Dec 2016

Although the authors' similarity report found some similarity with other papers by the authors, my main concern was a lack of similarity with one of them! Indeed, the authors completely failed to cite the first and still the most comprehensive multifractal analysis of satellite derived vegetation indices. Without this, their own paper is without adequate context, their results are simply isolated numbers – and as I argue – the numbers that are kept are stochastic variables and hence will lack reproduceability. This failure is remarkable because the second author of the paper was a key author in the earlier more thorough and quantitative one the same subject.

Please also note the supplement to this comment:

[Figure]

http://www.nonlin-processes-geophys-discuss.net/npg-2016-33/npg-2016-33-RC2-supplement.pdf

[Figure]

**Supplement:**

**Comments on: Spatial and radiometric characterization of multi-spectrum satellite imagesthrough multifractal analysis**

Carmelo Alonso, Ana M. Tarquis, Ignacio Zúñiga and Rosa M. Benito

Although the authors' similarity report found some similarity with other papers by the authors, my main concern was a lack of similarity with one of them! Indeed, the authors completely failed to cite the first and still the most comprehensive multifractal analysis of satellite derived vegetation indices. Without this, their own paper is without adequate context, their results are simply isolated numbers – and as we argue – the numbers that are kept are stochastic variables and hence will lack reproduceability. This failure is remarkable because the second author of the paper was a key author in the earlier more thorough and quantitative one the same subject.

In this paper, the authors use the multifractal *dimension* formalism of Halsey el 1986 that was developed for characterizing the deterministic phase spaces of strange attractors. In [*Lovejoy  et al*., 2008] co-authored by the second author in the present paper A. Tarquis (henceforth the "Tarquis paper"), it was explained in considerable detail why the dimension formalism is ill-suited for stochastic multifractals. Here, the images are assumed to be densities of multifractal measures, each realizations of a stochastic process. Co-author A. Tarquis can surely explain why the co-dimension formalism is more appropriate for the present application. She can also explain why the assumption of the existence of Holder exponents does not generally hold for stochastic multifractals and how the co-dimension formalism avoids this unnecessary (and doubtfully valid) assumption.

At the very least the paper must acknowledge the existence of the co-dimension formalism and refer to the Tarquis paper. The authors should also give the formulae:

$$f(\alpha) = d - c(\gamma); \quad \alpha = d - \gamma$$

$$\tau(q) = d(q-1) - K(q)$$

where *d* is the dimension of space (here *d* =2) and *c(γ)* is the codimension of the singularity of the density of the multifractal measure γ (γ is related to the singularity of the measure $\alpha$ by the formula $\alpha$ = d - γ above) and *K* is the moment scaling function of the density of the multifractal measure (i.e. it directly characterizes the scaling of the moments of the image rather than the integral of the image). These formulae are necessary in order to compare results obtained in the two formalisms (i.e. with the rest of the literature).

One of the advantages of the codimension formalism is immediately obvious from the formulae: c(γ), K(q) are independent of the dimension of the embedding space *d* whereas f($\alpha$), $\tau$(q) are different where ever one looks at subpaces of the

process (i.e. the same process but observed at different $d$). An related advantage of the codimension formalism is that when one performs the moment analysis (e.g. their figs 3, 6) that the moments will not dominated by the trivial, deterministic scaling factor $l^{d(q-1)}$ but will directly show the key (and usually much smaller) $l^{-K(q)}$ part (see the expression above; such an analysis is called "trace moment analysis"). As it is, the quality of the scaling of the statistics is practically impossible to judge from the authors' figures.

In addition - also as explained in the Tarquis paper - the moments $q<0$ will in general diverge so that special care is needed to avoid spurious estimates.

As carefully explained in the Tarquis paper, the multifractal spectrum f($\alpha$) - or better, c($\gamma$) - is a function; empirically it corresponds to estimating an infinite number of parameters. Since the framework is of stochastic processes, and in general stochastic multifractals have unbounded spectra (i.e. c($\gamma$) is generally unbounded), the authors differences $\Delta_{\pm}$ are simply random variables, they will provide very poor characterizations of the process. Why don't the authors characterize the multifractality as explained in the Tarquis paper (using $C_1$, and the multifractal index $\alpha$ - not the same as the authors' $\alpha$)? An added bonus would be that they could quantitatively compare their results with others in the literature (including those in the Tarquis paper!), rather than simply obtaining an isolated result with no context, no point of comparison. There are other ways of quantitatively characterizing the multifractality, but the singularity range used here is a particularly poor choice.

Another problem with the authors' characterization technique is that it ignores the issue of multifractal phase transitions that is extensively dealt with in the Tarquis paper. The authors should check that their moments (up to the rather high value of $q = 5$) are not spurious.

Some of the other conclusions of the Tarquis paper could also be recalled and the authors' new results could be then be quantitatively compared.

**Conclusion:** This paper should not be published without proper citations and comparisons with the Tarquis paper.

**Detailed Comments**:

Section 2.2, line 2: The authors state:
"A multifractal analysis is basically the measurement of a statistic distribution and therefore gives useful information even if the underlying structure does not show a full self similar behaviour (Plotnick et al., 1996). "

This is incomprehensible since isotropic multifractals assumed to be self-similar (i.e. scaling and isotropic), and the authors do not consider anisotropy in this paper. It is more correct to say that:

"A multifractal analysis is an analysis of how the statistical properties of a scaling field (or series) varies with scale. It therefore does *not* give useful information when the underlying structure is not scaling."

**References:**

Lovejoy , S., A. Tarquis, H. Gaonac'h, and D. Schertzer (2008), Single and multiscale remote sensing techniques, multifractals and MODIS derived vegetation and soil moisture, *Vadose Zone J.*, *7*, 533-546 doi: doi 10.2136/vzj2007.0173.

---

## Author Comment (AC1) · 24 Jan 2017

Please see the file attached where you have your specific answers (Answer_Ref#1), the answers to all the referees (answer) and the new version of the document with the changes marked in yellow (Alonso-et-al_Jan_17_v4).

Thank you for your work and to help us to improve the manuscript.

Please also note the supplement to this comment:
http://www.nonlin-processes-geophys-discuss.net/npg-2016-33/npg-2016-33-AC1-supplement.zip

---

## Author Response (AR1)

**Interactive comments on "Spatial and radiometric characterization of multi-spectrum satellite images through multifractal analysis" by Carmelo Alonso et al.**

**Answer to SCI1**

**In my opinion this is a novel work related to multiscaling analysis of data cropped from satellite images.**

*Thank you so much for your interest in this work. We have followed your research in multifractals applied in Soil Science and we hope to collaborate with you in the future.*

**I would like to pay attention to the following:**
**a) In order to better illustrate similitude or differences in the visible (blue, green and red) and near-infrared wavelength I suggest characterizing asymmetry of the singularity spectrum by AI index (Xie et al., 2010).**

**Reference: Xie, S., Q. Cheng, X. Xing, Z. Bao, and Z. Chen. 2010. Geochemical multifractal distribution patterns in sediments from ordered streams. Geoderma 160:36-46.**

*We have calculated the AI index in the table and include the reference you kindly has point out to us. In the new version the AI is included in the tables and in the Results and Discussion.*

**b) I wonder if it would be worth checking multifractality of the Normalized Difference Vegetation Index (NDVI)**

*In the new version we have included NDVI and EVI indexes to be analyzed and extract the multifractal spectrum. The results and discussion on these indexes have been added as well as we calculated the AI also.*

**Interactive comments on "Spatial and radiometric characterization of multi-spectrum satellite images through multifractal analysis" by Carmelo Alonso et al.**

**Answer to Anonymous Referee #1**

This manuscript characterizes several bands of multi-spectral images obtained with different radiometric and spatial resolution by IKONOS-2 and LANDSAT-7 satellites by multifractal analysis. Three of the studied bands were in the visible wavelength spectrum (red, green and blue), while the fourth one was in the near-infrared region. IKONOS-2 images were taken with a radiometric resolution of 11 bits/pixel and the pixel size was 4m x 4m pixel size. LANDSAT-7 images were taken with a radiometric resolution of 8 bits/pixel and the pixel size was 30m x 30m. For IKONOS-2 images both, the original pixel code (11 bits) and a transformed pixel code (8 bits) have been considered. Then, the effects of spatial and radiometric resolution on several multifractal parameters were investigated. The rationale and the objectives exposed in the Introduction section are worthwhile and in general the work appears well justified. The main findings are the usefulness of multifractal parameters to: 1) assess different patterns of scaling heterogeneity and evenness of the studied bands, and 2) discriminate between bands with different spatial and radiometric resolution. In general, the paper is well written and organized, and represents an original contribution. The results are based in robust data analysis. This study also is compatible with the aims of Nonlinear Processes in Geophysics (NPG) and may fit well into the scope of the current special issue titled "Multifractal Analysis in Soil Systems". In my opinion, it should be acceptable for publication following minor revisions. Some specific comments are next provided.

Thank you so much for your comments. We appreciate the work and the time you have dedicated to this manuscript and we have followed all your suggestions improving the manuscript to be considered in this special issue.

- The IKONOS-2 image presented and analyzed for multifractality is square and consists of 2048 x 2048 pixels of 4 x 4 m. The original LANDSAT-7 image is rectangular and consists of 772 x 828 pixels of 30 x 30 m; however multifractal analysis was not performed on the rectangular image, but on a square of 512 x 512 pixels. In my opinion this should be considered. To be consistent, I suggest modifying Figure 5 for including only the portion that has been used for multifractal analysis.

We have changed Figure 5 including only the part that was study. In this version Figure 5 is now Figure 6 as we have added more graphics for NDVI and EVI study.

- In general the discussion section should be tightened up. The authors should put an emphasis on what are the novel results and the novel things they have learned using multifractal analysis.

**We have improved the discussion section with a more detailed description of the results as you can see in the new version of the manuscript.**

 - Specifically, I wonder if the authors could include data about the effect of spatial and radiometric resolution on the vegetation indices mentioned in the Introduction, i.e. NDVI (normalized difference vegetative index) and EVI (enhanced vegetation index). Are changes in multifractal parameters related changes in NDVI, EVI or to other physical properties measured or derived from multispectral images?

**In the new version we have included NDVI and EVI indexes to be analyzed and extract the multifractal spectrum. The results and discussion on these indexes have been added.**

 - Tables 1, 2 and 3 list Hölder exponent parameters obtained from singularity spectra for q=1, q=2 and q=3. Please provide also the respective errors of these values.

**In these tables what we put is the singularity spectra for q=0, 1 and 2. We don't include q=3. In Tables 1, 2 and 3 the standard errors of the singularity for q=0, 1 and 2 have been included.**

 - I disagree with the statement in Page 8, Lines 13 and 14: "In order to avoid any other effect beside the spatial resolution a comparison between Landsat (with an original pixel code of 8 bits) and the rescaled histograms from Ikonos is made". This is because there are other factors of variation including 1) the total size of the image analyzed and 2) difference in the wavelength of a given spectral band between satellite images. Regarding my first remark, please note that if you change the size of the image, proportions of different vegetation and soil types are also changing and this has effects in the multispectral results. Regarding to the second remark, please note for example that wavelength for ETM#3 of Landsat was from 0.53 to 0.61 micrometers, while those for the same band of Ikonos was from 0.506 to 0.595.

**We agree with the reviewer and we have suppressed that phrase. Now it is as follows:**
To study the influence of radiometric resolution on Ikonos-2 image information complexity, the original pixel code (11 bits) has been transform to 8 bits through a rescaling based on minimum and maximum values between 0 and 255, with the aim of preserving the initial histogram shape.

- Even if the paper is reasonably well written, as before stated, the English language should be improved before acceptation.

**We apologize for this inconvenient. A native speaker has reviewed the English of this new version. We will pass it to a professional for the next version as we didn't want to delay more our answer.**

**Interactive comments on "Spatial and radiometric characterization of multi-spectrum satellite images through multifractal analysis" by Carmelo Alonso et al.**

**Answer to Anonymous Referee #2**

Although the authors' similarity report found some similarity with other papers by the authors, my main concern was a lack of similarity with one of them! Indeed, the authors completely failed to cite the first and still the most comprehensive multifractal analysis of satellite derived vegetation indices. Without this, their own paper is without adequate context, their results are simply isolated numbers – and as we argue – the numbers that are kept are stochastic variables and hence will lack reproduce ability. This failure is remarkable because the second author of the paper was a key author in the earlier more thorough and quantitative one the same subject.

Apologize for this mistake. We are totally agreed that we should include the paper you mention and the second author specially apologize for this mistake. The new version of this manuscript has included now the reference you mention:

Lovejoy , S., A. Tarquis, H. Gaonac'h, and D. Schertzer (2008), Single and multiscale remote sensing techniques, multifractals and MODIS derived vegetation and soil moisture, *Vadose Zone J.*, *7*, 533-546 doi: doi 10.2136/vzj2007.0173.

We had very few time to deliver this manuscript to be included in this special issue. We really appreciate to the editors the opportunity they gave us.

As we normally do in scientific context, each time that you refer to this paper, we will mention henceforth as Lovejoy paper, as Shaun Lovejoy is the first author of this paper.

In this paper, the authors use the multifractal *dimension* formalism of Halsey el 1986 that was developed for characterizing the deterministic phase spaces of strange attractors. In [*Lovejoy et al.*, 2008] co-authored by the second author in the present paper A. Tarquis (henceforth the "Lovejoy paper"), it was explained in considerable detail why the dimension formalism is ill-suited for stochastic multifractals. Here, the images are assumed to be densities of multifractal measures, each realizations of a stochastic process. Co-author A. Tarquis can surely explain why the co-dimension formalism is more appropriate for the present application. She can also explain why the assumption of the existence of Holder exponents does not generally hold for stochastic multifractals and how the co-dimension formalism avoids this unnecessary (and doubt fully valid) assumption.

The authors are working already on it to dedicate another manuscript to make a comparison of both methodologies on these images. This is an important issue that deserve a manuscript just with this aim, as it has been done in Morato et al. and Renosh et al. papers:

Morató, M.C., M.T. Castellanos, N.R. Bird, A.M. Tarquis. Multifractal analysis in soil properties: Spatial signal versus mass distribution. Geoderma, 287, 54-65, 2017. http://dx.doi.org/10.1016/j.geoderma.2016.08.004.

Renosh, P. R., Schmitt, F. G., and Loisel, H. 2015. Scaling analysis of ocean surface turbulent heterogeneities from satellite remote sensing: use of 2D structure functions. PLoS ONE, 10, e0126975. doi:10.1371/journal.pone.0126975.

**At the very least the paper must acknowledge the existence of the codimension formalism and refer to the Lovejoy paper. The authors should also give the formulae:**

$$f(\alpha) = d - c(\gamma); \quad \alpha = d - \gamma$$
$$\tau(q) = d(q-1) - K(q)$$

**where *d* is the dimension of space (here *d* =2) and *c(γ)* is the codimension of the singularity of the density of the multifractal measure *γ* (*γ* is related to the singularity of the measure α by the formula $\alpha = d - \gamma$ above) and *K(q)* is the moment scaling function of the density of the multifractal measure (i.e. it directly characterizes the scaling of the moments of the image rather than the integral of the image). These formulae are necessary in order to compare results obtained in the two formalisms (i.e. with the rest of the literature).**

**Now we have included in Material and Methods the followed in the subsection of Multifractal analysis:**

A monofractal object can be measured by counting the number N of δ size boxes needed to cover the object. The measure depends on the box size as

$$N(\delta) \propto \delta^{-D_0} \quad\quad\quad (3)$$

where

$$D_0 = \lim_{\delta \to 0} \frac{\log N(\delta)}{\log \frac{1}{\delta}} \quad\quad\quad (4)$$

[revised manuscript text omitted]

The width of the MF spectrum ($\Delta$) indicates overall variability (Tarquis et al., 2001; 2014) and we have split it in two sections. Section I correspond to values $\alpha(q) < \alpha(0)$ or $q > 0$ and section II to values with $\alpha(q) > \alpha(0)$ or $q < 0$. In section I the amplitude, or semi-width, was calculated with differences $\Delta^+ = \alpha(0) - \alpha(+5)$, and in section II with $\Delta^- = \alpha(-5) - \alpha(0)$.

To study the asymmetry of the multifractal spectrum we have choose the asymmetry index (AI) estimated as (Xie et al., 2010):

$$AI = \frac{\Delta\alpha_L - \Delta\alpha_R}{\Delta\alpha_L + \Delta\alpha_R} \qquad \begin{aligned} \Delta\alpha_L &= \alpha_0 - \alpha_{min} \\ \Delta\alpha_R &= \alpha_{max} - \alpha_0 \end{aligned} \qquad (10)$$

In our case, $\alpha_0$ is the singularity for q=0 or $\alpha(0)$, $\alpha_{min}$ is $\alpha(+5)$ and $\alpha_{max}$ is $\alpha(-5)$. Therefore, we can rewrite $AI$ as:

$$AI = \frac{\Delta^+ - \Delta^-}{\Delta^+ + \Delta^-} \qquad (11)$$

Expressing $AI$ as equation (11), we can see that it is a normalized index based on the amplitudes $\Delta^+$ and $\Delta^-$.

There are several works relating the UM model and the multifractal formalism based on $\tau(q)$ (Gagnon et al., 2003; Aguado et al., 2014; Morató et al., 2017 among others) through the equations:

$$f(\alpha) = E - c(\gamma); \ \alpha = E - \gamma \qquad (12)$$

$$\tau(q) = E(q-1) - K(q) \qquad (13)$$

where $E$ is the Euclidean dimension where the measure is embedded, in this case will be $E=2$, and $c(\gamma)$ is the codimension of the singularity of the density of the multifractal measure $\gamma$.

**One of the advantages of the codimension formalism is immediately obvious from the formulae: c($\gamma$), K(q) are independent of the dimension of the embedding space *d* whereas f($\alpha$), $\tau(q)$ are different where ever one looks at subpaces of the process (i.e. the same process but observed at different *d*). An related advantage of the codimension formalism is that when one performs the moment analysis (e.g. their figs 3, 6) that the moments will not dominated by the trivial, deterministic scaling factor $l^{d(q-1)}$ but will directly show the key (and usually much smaller) $l^{K(q)}$ part see the expression above; such an analysis is called "trace moment analysis"). As it is, the quality of the scaling of the statistics is practically impossible to judge from the authors' figures. In addition - also as explained in the Lovejoy paper – the moments *q<0* will in general diverge so that special care is needed to avoid spurious estimates.**

As carefully explained in the Lovejoy paper, the multifractal spectrum f($\alpha$) – or better, c($\gamma$) - is a function; empirically it corresponds to estimating an infinite number of parameters. Since the framework is of stochastic processes, and in general stochastic multifractals have unbounded spectra (i.e. c($\gamma$) is generally unbounded), the authors differences $\Delta\pm$ are simply random variables, they will provide very poor characterizations of the process. Why don't the authors characterize the multifractality as explained in the Lovejoy paper (using C1, and the multifractal index $\alpha$ - not the same as the authors' $\alpha$)? An added bonus would be that they could quantitatively compare their results with others in the literature (including those in the Lovejoy paper!), rather than simply obtaining an isolated result with no context, no point of comparison. There are other ways of quantitatively characterizing the multifractality, but the singularity range used here is a particularly poor choice.

We understand that you prefer the UM model than the multifractal spectrum and perhaps you consider the later a poor choice. However, the results are similar than the one found in Lovejoy paper. A quantitative comparison of both methodologies it will be the aim of our next manuscript where we can study deeper why discrepancies or agreements as it has been done in Morato et al. (2017) paper on a transect data of soil properties and Renosh et al. (2015) work applied on 2D remote sensing images. We appreciate these comments that will help us to improve the discussion in this next manuscript. Also it will be interesting to compare with the Structure Function and Detrended Fluctuation Analysis, other methods that we haven't mentioned here.

As mentioned in Morato et al. (2017) work introduction, the methodology we have applied here is the most common used in Soil Science for several reasons, and that is why we began to use it in this manuscript. Just looking into the NPG journal we can found several articles with this methodology used.

We agree that we shouldn't stop here and applied other type of methodologies that could be more interesting. We have added the follow at the end of Conclusions:

"Further research will be conducted to establish a qualitative and quantitative comparison of these conclusions among several multifractal methodologies applied on these images."

Lovejoy paper already included. Thanks so much to help us to avoid this mistake.